# HER2 Directed Antibody-Drug-Conjugates beyond T-DM1 in Breast Cancer

**DOI:** 10.3390/ijms20051115

**Published:** 2019-03-05

**Authors:** Gabriel Rinnerthaler, Simon Peter Gampenrieder, Richard Greil

**Affiliations:** 1Department of Internal Medicine III with Haematology, Medical Oncology, Haemostaseology, Infectiology and Rheumatology, Oncologic Center, Salzburg Cancer Research Institute—Laboratory for Immunological and Molecular Cancer Research (SCRI-LIMCR), Paracelsus Medical University Salzburg, 5020 Salzburg, Austria; g.rinnerthaler@salk.at (G.R.); s.gampenrieder@salk.at (S.P.G.); 2Cancer Cluster Salzburg, 5020 Salzburg, Austria

**Keywords:** ADC, HM2-MMAE, (vic-)trastuzumab duocarmazine, Trastuzumab deruxtecan, TAK-522, Trastuzumab emtansine, anti-HER2/PBD-MA, HER2 low, mode of action

## Abstract

Since the discovery of the human epidermal growth factor receptor 2 (HER2) as an oncogenic driver in a subset of breast cancers and the development of HER2 directed therapies, the prognosis of *HER2* amplified breast cancers has improved meaningfully. Next to monoclonal anti-HER2 antibodies and tyrosine kinase inhibitors, the antibody-drug conjugate T-DM1 is a pillar of targeted treatment of advanced HER2-positive breast cancers. Currently, several HER2 directed antibody-drug conjugates are under clinical investigation for *HER2* amplified but also HER2 expressing but not amplified breast tumors. In this article, we review the current preclinical and clinical evidence of the investigational drugs A166, ALT-P7, ARX788, DHES0815A, DS-8201a, RC48, SYD985, MEDI4276 and XMT-1522.

## 1. Introduction

The human epidermal growth factor receptor 2 (HER2), known as erbB-2, or proto-oncogene Neu, is a receptor tyrosine-protein kinase encoded by the *ERBB2 (HER2)* gene on chromosome 17q12 [1]. Besides epidermal growth factor receptor (EGFR, erbB-1), human epidermal growth factor receptor 3 (HER3, erbB-3), and human epidermal growth factor receptor 4 (HER4, erbB-4), HER2 is a member of the epidermal growths factor (EGF) receptor family. Since the HER2 protein has no ligand binding extracellular domain, no growth factors can directly bind to it. However, it forms heterodimers with ligand-binding members of the EGF receptor family, stabilizing ligand binding and enhancing kinase-mediated downstream signaling, including activation of phosphatidylinositol-3 kinase and mitogen-activated protein kinase [1,2].

HER2 expression can be detected on cell membranes of epithelial cells in the gastro-intestinal tract, respiratory tract, reproductive tract, urinary tract, skin, breast and placenta, but also on heart and skeletal muscle cells [3,4]. In fetal tissue, the level of HER2 expression is generally higher than in corresponding normal adult tissue [4].

A *HER2* amplification can promote tumorigenesis through multiple mechanisms and can therefore be considered as an oncogenic driver in *HER2* amplified cancers [1]. Besides breast cancer, *HER2* was found to be amplified and/or overexpressed in several cancer types including gastric and lung cancer [5].

Approximately 15% of all breast cancer cases belong to the HER2-positive subtype defined by HER2 protein overexpression and/or *HER2* gene amplification [6]. Traditionally, HER2-positive breast cancer was regarded as the most aggressive subtype and a high rate of recurrences were observed before the introduction of anti-HER2 targeted therapies. The addition of trastuzumab, a humanized monoclonal antibody targeting HER2, to conventional adjuvant chemotherapy, however, resulted in a significant and clinically relevant prolongation of disease free survival (HR 0.60; 95% confidence interval (CI) 0.50–0.71, *p* < 0.001) and overall survival (HR 0.66; 95% CI 0.57–0.77, *p* < 0.00001) [7]. Apart from trastuzumab, several other HER2-directed drugs such as the monoclonal antibody pertuzumab, the antibody-drug conjugate (ADC) trastuzumab-emtansine (T-DM1) and tyrosine-kinase inhibitors such as lapatinib and neratinib have found their way into the clinic, allowing targeted combination therapy or sequential administration of non-cross resistant drugs [8].

In about 50% of breast cancers a low-level expression of HER2 without HER2 amplification can be observed [9,10]. In two landmark adjuvant trastuzumab trials including patients with *HER2*-amplified or overexpressing breast cancer according to local site laboratories, a cohort of patients with neither HER2-amplification nor HER2 overexpression by central testing was identified. These HER2-low cohorts seemed to benefit from trastuzumab in a retrospective unplanned subgroup analysis [11,12]. The efficacy of an adjuvant trastuzumab treatment in HER2-low (immunohistochemistry (IHC) 1+ or 2+ but not *HER2* amplified) breast cancer patients was prospectively investigated in the phase 3 trial NSABP B-47 [13]. In this trial, 3270 patients were randomized 1:1 to standard adjuvant chemotherapy with or without one year of trastuzumab. No difference was observed between treatment groups concerning 5-year disease-free survival (DFS). The findings did not differ according by HER2 IHC level, extent of lymph node involvement, or hormone receptor status [13]. Despite HER2 amplification as a predictor for trastuzumab benefit, we recently demonstrated that a poly-ligand profiling can differentiate trastuzumab-treated breast cancer patients according to their outcomes [14].

Antibody–drug conjugates (ADCs) are molecules consisting of a recombinant monoclonal antibody covalently bound to a cytotoxic drug (called drug payload or warheads) via a synthetic linker [15]. ADCs combine the advantage of antibodies in binding a specific target and the cytotoxic capability of a chemotherapeutic drug. A stable linker between the antibody and the cytotoxic drug is crucial for the ADC integrity in circulation. After antibody binding to the specific antigen on the (cancer) cell surface, the ADC gets internalized and the cytotoxic drug is released intracellularly where it can exert its effect. Using cleavable linkers, ADCs can be designed to promote drug release from the target cell to the extracellular space. Thereby, surrounding and bystander cells, which may or may not express the ADC target antigen, can be killed by taking up the cytotoxic drug [15,16]. This bystander killing can also occur if the cytotoxic drug is released from the antibody after antigen binding just before internalization. The supposed mode of action of ADCs in HER2-low breast cancer patients is outlined in Figure 1.

T-DM1 is at present the only approved ADC for treatment of advanced HER2-positive breast cancer, based on the phase 3 registration trials EMILIA [17] and THERESA [18] comparing T-DM1 with capecitabine plus lapatinib and treatment of physicians choice, respectively. Recently, results of the phase 3 trial KATHRINE, where adjuvant T-DM1 was compared to trastuzumab in HER2-positive patients with residual disease after neoadjuvant chemo and anti-HER2 treatment, were published [19]. Due to the favorable efficacy of T-DM1, an approval in the post-neoadjuvant setting is awaited in 2019.

T-DM1 is a second-generation ADC consisting of the monoclonal HER2 directed IgG1 antibody trastuzumab, a non-cleavable thioether linker attached to random lysins and 3 to 4 maytansoinoid emtansine, also called DM1 [14]. In vitro, DM1 has a 11× to 25× higher cytotoxic potency than maytansine and is 24× to 270× more effective than taxanes. The mean ratio of DM1 molecules per antibody (drug antibody ratio, DAR) is 3.5 [20].

Currently, several ADCs are under clinical investigation for breast cancer treatment. Most of the drugs target HER2, but also other receptors like HER3, the zinc transporter LIV1, receptor tyrosine kinase-like orphan receptor 2 (ROR2) and Trop-2 serve as targets for the investigational drugs (Table 1 and Table 2). In this article we review the current evidence of investigational HER2 directed ADCs for the treatment of breast cancer.

## 2. A166

### 2.1. ADC Constituents

The ADC A166 is composed of a monoclonal anti-HER2 antibody conjugated to a cytotoxic agent. The monoclonal antibody and the cytotoxic agent have not been disclosed so far [21].

### 2.2. Ongoing Trials without Published Results

A166 is currently investigated in a running phase 1/2 trial including patients with relapsed or refractory HER2 expressing or *HER2* amplified cancers including breast cancer patients (clinicaltrials.gov identifier: NCT03602079). After defining the maximum tolerated dose (MTD) in the phase 1 dose escalation part of the trial, patients will be enrolled into several cohorts including HER2 positive breast cancer patients (cohort 1) and HER2-low breast cancer patients (IHC 1+ or 2+ but not *HER2* amplified; escalation cohort 3).

To the best of our knowledge, no published data of A166 are currently available.

## 3. ALT-P7 (HM2/MMAE)

### 3.1. ADC Constituents

ALT-P7 is an ADC composed of the trastuzumab biobetter HM2 conjugated in a site-specific manner to monomethyl auristatin E (MMAE) [22]. MMAE is a cytotoxic agent acting that inhibits the tubulin polymerization in dividing cells resulting in a G2/M phase arrest and apoptosis [23].

### 3.2. Ongoing Trials without Published Results

ALT-P7 is currently investigated in an ongoing open-label, dose escalation and phase 1 trial in patients with HER2 positive MBC, who have progressed on previous trastuzumab-based therapy (clinicaltrials.gov identifier: NCT03281824).

To our knowledge, no published data of ALT-P7 are currently available.

## 4. ARX-788 (ARX788)

### 4.1. ADC Constituents

The novel ADC ARX-788 is composed of a monoclonal HER2 targeting antibody site-specifically conjugated, via a non-natural amino acid linker para-acetyl-phenylalanine (pAcF), to monomethyl auristatin F (MMAF) [24]. The site-specific conjugation of MMAF to the HER2 antibody improves the therapeutic window of ARX-788 by increasing payload stability and optimizing its half-life. The mean DAR is 1.9.

### 4.2. Preclinical Data

In murine xenograft models of the HER2-positive breast cancer cell lines BT474 and HCC1954, rapid tumor regression was induced after a single injection of ARX-788 [24]. In a trastuzumab-resistant breast cancer xenograft model (JIMT-1), ARX-788 was significantly more effective than T-DM1 in inducing tumor regression. Long-term stability of 12 and 8 days was achieved in rodent and non-human primates, respectively. The conjugated form of ARX-788 remained intact over the course of a 3-week study in non-human primates.

### 4.3. Ongoing Trials without Published Results

ARX-788 is currently investigated in two ongoing phase 1 trials (clinicaltrials.gov identifiers: NCT02512237 and NCT03255070). The aim of the first parts of the trials (phase 1a) is to determine the recommended phase 2 dose (RP2D) in patients with HER2 positive advanced solid tumors. In the second part (phase 1b) of the first trial, safety and activity of the RP2D will be tested in three expansion cohorts: a HER2-positive advanced breast cancer cohort, a HER2-low advanced breast cancer cohort, and a HER2 positive gastric cancer cohort (clinicaltrials.gov identifier: NCT02512237). Phase 1b of the second trial is designed to investigate the activity and safety of the RP2D in two advanced breast cancer expansion cohorts: one cohort with HER2 positive patients, and one cohort with HER2-low patients (clinicaltrials.gov identifier: NCT03255070).

## 5. DHES0815A (Anti-HER2/PBD-MA)

### 5.1. ADC Constituents

DHES0815A consists of a monoclonal HER-2 targeting antibody linked to pyrrolo[2,1-c][1,4]benzodiazepine monoamide (PBD-MA) [25]. PBD-MA crosslinks DNA minor grooves, leading to DNA strand breaks, cell cycle arrest, and cell death.

### 5.2. Ongoing Trials without Published Results

DHES0815A is currently investigated in a first-in-human, open-label, multicenter, dose-escalation phase 1 trial evaluating the safety, tolerability, and pharmacokinetics (clinicaltrials.gov identifier: NCT03451162).

To our knowledge, no published data of DHES0815A are currently available.

## 6. DS-8201a (Trastuzumab Deruxtecan)

### 6.1. ADC Constituents

DS-8201a is a novel ADC composed of trastuzumab, an enzymatically cleavable maleimide glycynglycyn-phenylalanyn-glycyn (GGFG) peptide linker and a topoisomerase I inhibitor [26]. Topoisomerase I inhibitors induce double-strand DNA breaks and apoptosis by binding to and stabilization of topoisomerase I-DNA cleavable complexes [27]. DXd, the topoisomerase I inhibitor component of DS-8201a, is a derivative of exatecan mesylate (DX-8951f). In various tumor xenograft models, including CPT-11–resistant tumors, antitumor activity of DX-895 was superior to irinotecan (CPT-11) [28]. Each trastuzumab molecule of DS-8201a is conjugated with 8 molecules of DXd [26]. This DAR of 8 is higher compared to T-DM1 with a DAR of 3-4 [20]. After binding to HER2 on the cell surface, DS-8201a gets internalized and the linker is cleaved by lysosomal enzymes such as cathepsins B and L which are highly expressed in tumor cells [26].

### 6.2. Preclinical Data

A potent bystander effect of DS-8201a is suggested due to a high membrane-permeability of the DS-8201a payload DXd [29]. In comparison, the payload of T-DM1, Lys-SMCC-DM1, has a low level of permeability. In coculture experiments with HER2-positive KPL-4 cells and HER2-negative MDA-MB-468 cells, DS-8201a killed both cells lines, whereas T-DM1 could not. This observation was confirmed in a xenograft model [15].

In various mice xenograft models with different HER2 expression levels, DS-8201a proved also effective in moderate HER2 positive and weak HER2 positive models, while T-DM1 efficacy was limited to a strong HER2 positive model [29]. In HER2-low patient derived xenografts (PDX), an antitumor activity of DS-8201a could be shown, which was not the case with T-DM1.

### 6.3. Clinical Data

Single agent DS-8201a is currently investigated in a large phase 1 trial in heavily pretreated patients with HER2 expressing solid tumors, including patients with breast cancer (clinicaltrials.gov identifier NCT02564900) [30,31]. Twenty-four patients were enrolled into the dose escalation part (part 1), and a further 260 patients will be enrolled into several dose expansion cohorts (part 2). In part 1, DS-8201a was administered at a dose of up to 8.0 mg/kg. No dose limiting toxicity was observed and the maximum tolerated dose (MTD) was not reached. For part 2, dose levels of 6.4 and 5.4 mg/kg IV every 3 weeks were chosen [30].

In 99% of patients treated with 5.4 or 6.4. mg/kg (*n* = 241, data cutoff April 2018), an adverse event (AE) of any grade was observed [31]. AEs ≥ grade 3 occurred in 42% of patients and serious adverse events were reported in 21% of patients. The most common non-hematological AEs were nausea (all grades: 69%, grade ≥ 3: 3%), vomiting (all grades: 35%, grade ≥ 3: 2%), diarrhea (all grades: 27%, grade ≥ 3: 1%), decreased appetite (all grades: 56%, grade ≥ 3: 3%), alopecia (all grades: 36%) and fatigue (all grades: 28%, grade ≥ 3: 2%). Anemia (all grades: 32%, grade ≥ 3: 15%), thrombocytopenia (all grades: 29%, grade ≥ 3: 10%) and neutropenia (all grades: 25%, grade ≥ 3: 15%) were common. The frequency of infusion-related reactions (all grades: 2%, grade ≥ 3: 0%) was low. Laboratory abnormalities of liver enzymes were generally of low grade (AST increase: all grades: 20%, grade ≥ 3: 1%; ALT increase: all grades: 16%, grade ≥ 3: 1%). A decrease in ejection fraction (all grades: 1%, grade ≥ 3: 0%) and a QT prolongation (all grades: 5%, grade ≥ 3: <1%) were rarely observed. Interstitial lung disease (ILD; all grades: 3%, grade ≥ 3: 1%) and pneumonitis (all grades: 7%, grade ≥ 3: 2%) were infrequently doucmented, but 5 fatal cases were attributed to ILD and pneumonitis.

As of April 2018, 111 patients with HER2-positive metastatic breast cancer evaluable for efficacy outcome were enrolled in this phase 1 trial, with a median age of 55 (range 33–77) and a median of 7 prior therapies (range 2–21) [31]. The confirmed overall response rate (ORR) was 55% with a disease control rate (DCR: CR + PR + SD) of 94%. Median duration of response and median progression-free survival (PFS) were not reached. Confirmed ORRs at dose levels 5.4 and 6.4 mg/kg were 53% and 56%, respectively [31]. The pharmacokinetic relationship between minimum blood plasma concentration C_min_ of intact DS-8201a and ORR was statistically significant (*p* = 0.035). Based on logistic regression, a statistically significant relationship was observed between applied dose and the following AEs: neutropenia (any grade, *p* = 0.003; grade ≥ 3, *p* = 0.037), anemia (any grade, *p* = 0.002; grade ≥ 3, *p* < 0.001), thrombocytopenia (any grade, *p* = 0.021), ILD/pneumonitis (any grade, *p* = 0.017), but also dose reduction due to AE (*p* = 0.003) and discontinuations because of AEs (*p* = 0.035). Additionally, Cox proportional hazard modeling suggested a higher risk of ILD with higher exposure of intact DS-8201a (any grade, *p* < 0.001; grade ≥ 2, *p* = 0.007). Based on the predicted benefit / risk profile, 5.4mg/kg DS-8201a was chosen as the recommended dose for further development of DS-8201a in HER2-positive breast cancer.

Thirty-four patients with HER2-low breast cancer were enrolled at data cutoff (12 October 2018), with a median age of 56 years (range 33–75) and a median number of prior cancer regimens of 8 (2–18) [32]. The majority of HER2-low patients had hormone-receptor positive disease (87%) and 34% of them were pretreated with a CDK4/6 inhibitor. The confirmed ORR was 44%, the DCR was 79% and the median time to response was 2.8 months (range 1.6–3.0 months). Median PFS was 7.6 months (95% CI 4.9–13.7) and duration of response (DOR) was 9.4 months (95% CI 1.5–23.6). In a subgroup analysis based on HER2 IHC expression, ORR (54% vs. 33%) and median PFS (13.6 months vs. 5.7 months) were superior in IHC 2+ tumors (*n* = 24) compared to IHC 1+ tumors (*n* = 27).

### 6.4. Ongoing Trials without Published Results

Currently seven registered trials investigating DS-8201a are active. In a phase 1b trial with a dose escalation and an expansion cohort, the combination of DS-8201a with the PD-1 checkpoint-inhibitor nivolumab in patients with breast and urothelial carcinomas has been studied (clinicaltrials.gov identifier: NCT03523572). One phase 1 trial assesses the safety and pharmacokinetics in HER2-positive advanced gastric cancers, gastroesophageal junction adenocarcinomas or breast cancers (clinicaltrials.gov identifier: NCT03368196). A third phase 1 trial investigates the effect on QT intervals and pharmacokinetics of different DS-8201a doses in patients with HER2-positive breast cancer (clinicaltrials.gov identifier: NCT03366428). In the phase 2 trial DESTINY-Breast01, HER2-positive breast cancer patients resistant, refractory or intolerant to T-DM1 are randomized into different DS-8201a dose level groups, to assess pharmacokinetics and recommended dose, followed by an expansion cohort (clinicaltrials.gov identifier: NCT03248492).

The randomized, open-label phase 3 trial DESTINY-Breast02 compares DS-8201a with physicians’ choice (trastuzumab plus capecitabine or lapatinib plus capecitabine) in patients with HER2-positive advanced breast cancer (ABC) pretreated with prior standard of care HER2 targeting therapies including T-DM1 (clinicaltrials.gov identifier: NCT03523585). The two ADCs DS-8201a and T-DM1 are compared in the randomized, open-label phase 3 trial DESTINY-Breast03 in pretreated patients with HER2-positive ABC (clinicaltrials.gov identifier: NCT03529110). The third ongoing randomized, open-label phase 3 trial (DESTINY-Breast04), investigates DS-8201a versus physicians’ choice (capecitabine, eribulin, gemcitabine, paclitaxel or nab-paclitaxel) in patients with HER2-low (IHC 1+ or IHC 2+/in situ hybridization (ISH)−) ABC (clinicaltrials.gov identifier: NCT03734029).

## 7. MEDI4276

### 7.1. ADC Constituents

MEDI4276 is a novel ADC composed of a HER2-bispecific antibody targeting two different epitopes on HER2, site-specifically conjugated via a maleimidocaproyl linker to the potent tubulysin-based microtubule inhibitor AZ13599185 [33,34]. The small-molecule toxin AZ13599185 inhibits microtubule polymerization during mitosis and thereby induces cell death [34]. The bispecific antibody contains four antigen-binding units, two on each arm that are capable of interacting with two different epitopes on HER2. Antibody interaction with the unique HER2 epitope, different to the trastuzumab and pertuzumab binding epitope, can completely interfere with HER2-HER3 receptor dimerization induced by heregulin-1β. Therefore, this bispecific antibody blocks both ligand-independent and ligand-dependent receptor activation. The DAR of MEDI4276 is 4.

### 7.2. Preclinical Data

In vitro, MEDI4276 was at least 10-fold more potent in the HER2 overexpressing cell line SKBR-3, than T-DM1 [33]. Additionally, MEDI4276 demonstrated efficacy in the T-DM1 resistant HER2 positive JIMT-1 cell line. In HER2-low cell lines (MCF7-GTU, and ZR-75-1), MEDI4276 demonstrated potent anti-tumor activity whereas T-DM1 was ineffective. In a patient derived HER2-positive breast cancer xenograft model, weekly intravenous administration of MEDI4276 over four weeks induced a complete remission in all treated animals and the animals remained tumor free over 120 days after treatment discontinuation. In contrast, T-DM1 only induced tumor stasis and a rapid regrowth was observed after T-DM1 treatment cessation. In several patient derived HER2-low xenograft models, MEDI4276 induced tumor regression regardless of the hormone receptor status.

### 7.3. Clinical Data

In a phase 1/2 dose escalation and dose expansion trial MEDI4276 was investigated in patients with advanced pretreated HER2 expressing (IHC 2+) breast or gastric cancer (clinicaltrials.gov identifier NCT02564900) [35]. As of November 2017, 43 patients had been enrolled and treated in several dose cohorts (0.05, 0.1, 0.2, 0.3, 0.4, 0.5, 0.6, 0.75, or 0.9 mg/kg every 3 weeks), following a 3+3 design. MTD was determined as 0.9 mg/kg. Drug-related AEs of any grade were reported in 88% of patients and AEs of grade ≥ 3 were reported in 12% of patients. The most common AEs were nausea (all grade: 58%), fatigue (all grade: 42%), elevated AST (all grade: 37%, grade ≥ 3: 19%), vomiting (all grade: 37%), and elevated ALT (all grade: 35%, grade ≥ 3: 12%). Drug-related peripheral neuropathy grade 3 was observed in 1 patient (2%) at 0.6 mg/kg and in 2 patients (5%) at 0.75 mg/kg. In evaluable patients 1 CR (0.5 mg/kg; breast cancer), 1 PR (0.6 mg/kg; breast cancer), and 12 SD (28%) were reported. A non-linear pharmacokinetic with rapid clearance and negligible deconjugation of MEDI4276 was observed.

The preclinical data and early clinical data of MEDI4276 support further clinical development of this drug in HER2-positive and HER2-low breast cancer patients.

## 8. RC48 (RC48-ACD, Hertuzumab-vc-MMAE)

### 8.1. ADC Constituents

RC48 is the novel humanized anti-HER2 antibody hertuzumab conjugated with monomethyl auristatin E (MMAE) via a cleavable linker [36]. MMAE acts by inhibiting the tubulin polymerization in dividing cells resulting in a G2/M phase arrest and apoptosis [23]. Hertuzumab had a higher affinity to HER2 than trastuzumab in an ELISA-based binding assay [36]. The monoclonal anti-HER2 antibody binds specifically to HER2, but not to other members of the human epidermal growth factor receptor family (EGFR, HER3, or HER4). MMAE is linked to hertuzumab using a protease-sensitive valine-citrulline dipeptide sequence, which was designed for optimal stability in human plasma and efficient cleavage by human cathepsin B. The DAR is approximately 4. After binding of RC48 to HER2 on the cell surface, the R48-HER2 complex is internalized through endocytosis. Following internalization, lysosomal proteases cleave both, the monoclonal antibody and the linker, and MMAE is released.

### 8.2. Preclinical Data

In vivo efficacy of RC48 was investigated in trastuzumab and lapatinib sensitive and resistant breast cancer xenograft models in female nude BALB/cA mice subcutaneously implanted with breast cancer cells [36]. For the trastuzumab and lapatinib sensitive model, BT-474 human breast cancer cells, which express high levels of HER2, were implanted. Antitumor activity of RC48 (0.5, 1.5, and 5.0 mg/kg) was dose-dependent. RC48 activity at dose levels ≥0.5 mg/kg was significantly enhanced compared to trastuzumab (10 mg/kg) and lapatinib (200 mg/kg). In the resistant breast cancer models, nude mice were implanted with BT-474/T721 (trastuzumab-resistant) and BT-474/L1.9 (trastuzumab- and lapatinib- resistant) cells, respectively. Both RC48 (5.0 mg/kg) and T-DM1 (5.0 mg/kg) showed higher effectivity in the BT-474/T721 xenograft model compared to trastuzumab. In the trastuzumab and lapatinib resistant BT474/L1.9 xenograft model, RC48 (5.0 mg/kg) was more effective than trastuzumab, lapatinib and T-DM1.

### 8.3. Clinical Data

RC48 is being investigated in a dose escalation open-label, single-center phase 1 trial in HER2-positive breast cancer patients (clinicaltrials.gov identifier: NCT02881138) [37]. As of January 2018, 23 patients have been treated in 5 dose escalation cohorts (dose levels 0.5, 1.0, 1.5, 2.0, 2.5 mg/kg) once every two weeks (Q2W) following a 3+3 design. Median age was 57 years (range 32–65), median number of prior treatment lines in the metastatic setting was 3 (range 1–6), and 70% (16/23) of patients had been pretreated with trastuzumab. MTD has not been reached at doses up to 2.0 mg/kg Q2W. The most common treatment-related adverse events (AEs) were leukopenia (all grades: 48%, grade ≥ 3: 4%), AST elevation (all grades: 48%, grade ≥ 3: 4%) and neutropenia (all grades: 43%, grade ≥ 3: 13%). In patients treated with doses ≥ 1.5 mg/kg (14 evaluable patients for response), ORR was 57% (8/14) and DCR was 86% (12/14). As MTD has not been determined at dose levels of up to 2.0 mg/kg twice-weekly, a 2.5 mg/kg Q2W dose escalation cohort is ongoing.

A second dose escalation phase 1 trial investigates RC48 in patients with HER2-overexpressing advanced solid cancers (clinicaltrials.gov identifier: NCT02881190) [38]. As of January 2018, 36 patients, including 1 breast cancer patient, have been enrolled in dose escalation (0.1–2.5 mg/kg Q2W and 2.0 mg/kg Q3W) and dose expansion cohorts, respectively. The most common treatment-related AEs were in line with the previously mentioned trial: AST elevation (all grades: 50%, grade ≥ 3: 3%), ALT elevation (all grades: 43%, grade ≥ 3: 3%), leukopenia (all grades: 33%, grade ≥ 3: 7%), neutropenia (all grades: 33%, grade ≥ 3: 10%) and numbness (all grades: 23%, grade ≥ 3: 0%). No AEs grade ≥ 4 were observed. Pharmacokinetic analyses demonstrated a dose-dependent exposure with a 1–1.5 days half-life [39]. A further expansion cohort investigating a dose of a 2.5 mg/kg is planned.

An open-label, multicenter phase 1b/2 trial investigates RC42 in patients with pretreated metastatic HER2-positive breast cancer (clinicaltrials.gov identifier: NCT03052634) [39]. As of January 2018, 30 patients (6 IHC 2+/ISH+; 24 IHC 3+) have been enrolled in 1.5 and 2.0 mg/kg cohorts in the phase 1b part of the trial. Median age was 53 years (range 26–62), 19 patients (63%) had been pretreated with HER2-targeting drugs and 16 patients (53%) had been pretreated with ≥3 prior chemotherapy regimens in the metastatic setting. ORR was 37% (11 PR) and DCR was 97% (29/30) with a clinical benefit rate (CBR; CR + PR + SD ≥ 6 months) of 47% (14/30). In the 1.5 mg/kg and 2.0 mg/kg cohorts the ORR was 27% and 47%, respectively. In trastuzumab-naive and trastuzumab-pretreated patients, ORR was 57% and 33%, respectively. Most common treatment-related AEs were in line with the two previously mentioned RC42 phase 1 trials: AST elevation (all grades: 50%, grade ≥ 3: 3%), ALT elevation (all grades: 43%, grade ≥ 3: 3%), leukopenia (all grades: 33%, grade ≥ 3: 7%), neutropenia (all grades: 33%, grade ≥ 3: 10%), numbness (all grades: 23%, grade ≥ 3: 0%). Thrombocytopenia (all ≤ grade 2) was observed in 10% of patients. No grade ≥ 4 AEs were observed. Enrollment in the 2.5 mg/kg expansion cohort is underway. In the planned phase 2 part of the trial, patients will be randomized to RC48 at the dose level selected in phase 1b or to lapatinib plus capecitabine.

### 8.4. Ongoing Trials without Published Results

A randomized, multicenter, 2-arm, open-label phase 2 trial comparing RC48 (2.0 mg/kg Q2W) with capecitabine plus lapatinib in trastuzumab pretreated patients with advanced HER2-positive breast cancer is currently recruiting in Chinese trial centers (clinicaltrials.gov identifier: NCT02881138).

The novel ADC RC42 demonstrated a favorable toxicity profile in three phase 1 trials. Response rates ranging between 37% and 57% in partly heavily pretreated patients with HER2-positive breast cancer patients are promising, and further clinical development of this drug is warranted.

## 9. SYD985 ([vic-]Trastuzumab Duocarmazine)

### 9.1. ADC Constituents

SYD985 is composed of the monoclonal HER2 directed antibody trastuzumab linked via a cleavable valine-citrulline peptide to the synthetic duocarmycin analogon seco-DUocarmycin-hydroxyBenzamide-Azaindole (vc-seco-DUBA) [40,41]. Duocarmycins are DNA-alkylating agents composed of a DNA-alkylating and a DNA-binding moiety binding to the minor groove of the DNA causing irreversible alkylation of DNA [41]. These cytotoxic drugs induce cell death in both dividing and nondividing cells by disrupting the nucleic acid architecture. The average DAR is 2.8 [40].

### 9.2. Preclinical Data

In a trastuzumab sensitive BT474 mouse xenograft model, antitumor activity of SYD985 was dose dependent [41]. Antitumor activity of 1 mg/kg SYD985 was equal to 5 mg/kg trastuzumab and SYD985 dosed once at 5 mg/kg significantly reduced tumor growth compared to trastuzumab at the same dose level. In HER2-positive (IHC 3+) breast cancer patient-derived xenograft models named MAXF1322 and MAXF1162, SYD985 dose dependently reduced tumor growth, whereas high dose trastuzumab did not show any antitumor activity. In a HER2-positive (IHC 3+) breast cancer cell line (SK-BR-3) and trastuzumab-resistant breast cancer cell line (UACC-893), SYD985 and T-DM1 demonstrated similar potencies [42]. In two HER2-low (HER 1+) cell lines (MDA-MB-175-VII and ZR-75-1), SYD985 retained its activity, whereas T-DM1 was less potent [42]. Neither SYD985 nor T-DM1 was able to kill HER2-negative cells (SW-620 and NCI-H520) [42]. These findings were confirmed in vivo where SYD985 was active in HER2-low breast cancer xenograft models, which was not the case for T-DM1 [42]. In coculture experiments of HER expressing cells (SK-BR-3 and MDA-MB-175-VII) with HER2 negative (HER2 0) NCI-H520 cells, a bystander killing was observed in the presence of SYD985, but not of T-DM1 [42].

### 9.3. Clinical Data

SYD985 was investigated in a two-part phase 1 trial (clinicaltrials.gov identifier: NCT02512237). In the dose-escalation part of the study, patients with solid tumors and any HER2 status (*n* = 39), including 26 patients with breast cancer, were enrolled and treated with SYD985 at doses varying from 0.3 to 2.4 mg/kg every three weeks [43]. The RP2D was defined as 1.2 mg/kg Q3W. Patients with HER2 expressing breast, gastric, urothelial or endometrial cancers were subsequently enrolled in expansion cohorts treated with the RP2D. Breast cancer patients (*n* = 26) enrolled in the dose-escalation part were heavily pretreated with a median of seven systemic therapies. All HER2-positive patients were pretreated with T-DM1. As of May 2016, tumor evaluation data were available for 19 of 26 enrolled breast cancer patients. In evaluable HER2-positive patients (*n* = 14), ORR was 36% (5/14) and DCR was 93% (13/14). In evaluable HER2-low patients (*n* = 5) an ORR of 60% (3/5) and a DCR of 80% (4/5) was observed. In evaluable patients treated with doses ≥1.2 mg/kg ORR was 42% and 75% for the HER2-positive and HER2-low patients, respectively. One fatal pneumonitis occurred at 2.4 mg/kg of SYD985. Up to doses of 1.8 mg/kg every 3 weeks, SYD985 was well tolerated. The most frequently reported drug-related AEs were conjunctivitis, stomatitis, fatigue, and decreased appetite and the majority of these AEs were of mild or moderate intensity.

Ninety-nine breast cancer patients were enrolled in dose expansion cohorts: 50 patients with HER2-positive MBC, 32 patients with HER2-low hormone-receptor positive disease and 17 patients with HER2-low triple negative MBC [44]. The median number of prior cancer regimens was 6 (range 1–21) and 80% of the HER2-positive patients were pretreated with T-DM1. In HER-positive patients ORR was 33% (16/48 patients with measurable disease) and PFS was 9.4 months (95% CI 4.5–12.4). In T-DM1 pretreated HER2 positive patients, an ORR of 29% (11/38) and a PFS of 8.3 months (95% CI 4.1–15.0) was observed. ORR was 27% (8/30) and 40% (6/15), and PFS was 4.1 (95% CI 2.4–5.4) and 4.4 (95% CI 1.0–7.1) in patients with HER2-low hormone-receptor positive and HER2-low triple negative disease, respectively. The most common drug related AEs in patients of all expansion cohorts (*n* = 146) were fatigue (all grades: 32%, grade ≥ 3: 3%), dry eyes (all grades: 29%, grade ≥ 3: 1%), conjunctivitis (all grades: 25%, grade ≥ 3: 3%) and nausea (all grades: 20%, grade ≥ 3: 0%). The majority of AEs were grade 1 or 2 whereas 6% were grade 3 AEs. No ≥ grade 4 AEs were observed. Twenty-eight (19%) patients discontinued treatment due to AEs, most commonly due to ocular toxicity. Alopecia was reported in 18% of patients (grade 1: 15%, grade 2: 3%).

### 9.4. Ongoing Trials without Published Results

SYD985 is currently investigated in a multi-center, open-label, randomized phase 3 trial comparing SYD985 with physicians’ choice in patients with HER2-positive advanced or metastatic breast cancer pretreated with T-DM1 (clinicaltrials.gov identifier: NCT03262935).

SYD985 was well tolerated and ocular toxicity was commonly reported in a large phase 1 trial. The efficacy data of a phase 1 expansion cohort in T-DM1 pretreated patients with HER2 positive breast cancer are promising. The results of an ongoing phase 3 trial in this patient population are awaited within the next two years.

## 10. XMT-1522 (TAK-522)

### 10.1. ADC Constituents

XMT-1522 is an ADC composed of a novel IgG1 anti-HER2 monoclonal antibody (HT-19) conjugated with the Dolaflexin^®^ platform to auristatin-based drug payload molecules (Auristatin F-hydroxypropylamide, AF-HPA) [45,46]. The Dolafexin^®^ platform is a biodegradable polymer-based conjugation platform that enables a high average XMT-1522 DAR of 12 (range 10–15) without aggregation or detrimental impact on pharmacokinetics [47]. Auristatin analogs act by inhibiting the tubulin polymerization in dividing cells resulting in a G2/M phase arrest and apoptosis [23]. The HT-19 antibody is non-competitive with trastuzumab or pertuzumab for HER2 binding [45].

### 10.2. Preclinical Data

Across a panel of 25 tumor cell lines with different HER2 expression levels, XMT-1522 was approximately hundred times more potent than T-DM1 [45]. In a BT-474 HER2-positive breast cancer xenograft model, a single dose of 5 mg/kg HT-19 antibody was ineffective, while a single dose of 2 mg/kg or 5 mg/kg XMT-1522 induced durable complete tumor regression, indicating that the primary mechanism of XMT-1522 is cytotoxic payload delivery, not HER2 signaling inhibition [45]. In the same model, T-DM1 at a single dose of 5 mg/kg was ineffective. In a patient-derived HER2-positive xenograft model, XMT-1522 induced durable complete tumor regression after a single 1 mg/kg dose, while a 10 mg/kg dose of T-DM1 resulted in tumor growth delay without regression [45]. In a patient-derived HER2-low xenograft model, XMT-1522 at a single 3 mg/kg dose achieved partial tumor regression, whereas T-DM1 was ineffective [45].

In vitro, a combination of XMT-1522 with trastuzumab did not block the XMT-1522 HER2 binding ability or the ADC internalization [45]. In a HER2 positive xenograft model, a combination of trastuzumab, pertuzumab and XMT-1522 was synergistic. Despite the high potency of XMT-1522 in HER2-low tumor models, no XMT-1522-related toxicity was observed in HER2-expressing tissues including heart and lung [48].

In multiple cell lines, an immunogenic cell death, as measured by cell surface expression of calreticulin, was induced a few hours after treatment with free AF-HPA and XMT-1522 [49]. In a HER2-low breast cancer (4T1) xenograft model, XMT-1522 but not T-DM1 significantly inhibited tumor growth [49]. A combination of XMT-1522 with an anti-PD1 monoclonal antibody synergistically enhanced the anti-tumor efficacy, with complete responses in some mice. The frequency of complete remissions was further enhanced when the two drugs were given sequentially (XMT-1522 followed by the checkpoint inhibitor).

### 10.3. Clinical Data

XMT-1522 is currently investigated in the first-in-human phase 1b dose escalation and expansion trial in patients with advanced HER2-expressing (IHC ≥ 1+) breast cancer, gastric cancer and non-small cell lung cancer progressing on standard therapy (clinicaltrials.gov identifier: NCT02952729). XMT-1522 is administered intravenously every 3 weeks. Dose escalation uses a 3+3 design and a 3-week dose limiting toxicity (DLT) evaluation period. As of February 2018, 19 patients have completed the DLT evaluation period across 6 dose levels (2 to 21.3 mg/m^2^ every 3 weeks). Since no DLT, no serious adverse event (SAE) and no treatment-related AE ≥grade 3 have been observed, dose escalation was continued [50]. The most common treatment-related AE were elevated liver enzymes, fatigue, nausea, vomiting, headache, and anorexia. ORR was 17% (1/5 evaluable patients) and DCR was 83% (5/6) in patients dosed at 16 or 21.3 mg/m^2^. The partial remission was observed at the first assessment in a patient with HER2-positive breast cancer previously treated with T-DM1. In patients treated at a dose of less than 16 mg/m^2^, DCR was 25% (3/12) without observed responses. Systemic exposure of total AF-HPA payload was approximately dose-proportional. Plasma concentrations of free AF-HPA and its active metabolites were low.

XMT-1522 has interesting biochemical features with a higher drug-antibody ratio and a novel HER2 antibody. Preclinical data and first clinical data are promising and the final results of the first-in-human phase 1 trial are awaited within the next year.

## 11. Discussion

Antibody-drug conjugates are a promising class of anti-cancer drugs combining the selectivity of monoclonal antibodies and the cell killing potential of cytotoxic agents [15,51]. Targeted cytotoxic drug delivery into tumor tissue increases the therapeutic window of these agents considerably. For example, clinical development of unconjugated DM1 was stopped early due to unfavorable toxicity despite promising clinical activity [51]. In contrast, T-DM1, consisting of DM1 attached via a non-cleavable linker to trastuzumab, has a favorable toxicity profile and a clinically meaningful antitumor activity in HER2 positive breast cancer [17].

HER2 is a rational target for ADCs in breast cancer as about 15% of breast cancers show HER2 protein overexpression and/or HER2 gene amplification [6], and a further 50% show a low-level expression of HER2 without HER2 amplification [9,10]. Additionally, HER2 testing in breast cancer is standardized by international guidelines and reporting of the HER2 status mandatory in clinical routine [9].

The published preclinical and clinical data of the reviewed investigational HER2 directed ADCs A166, ALT-P7, ARX788, DHES0815A, DS-8201a, RC48, SYD985, MEDI4276 and XMT-1522 are promising. In preclinical models, most of these drugs were more effective than T-DM1, which raises high expectations for these novel drugs. The investigation of SYD985 and DS-8201a in T-DM1-refractory HER2-positive patients in currently enrolling randomized phase 3 trials is straight forward.

Which of the plethora of new ADCs will find its way into the clinic remains speculative. Interestingly, the toxicity profile of the different compounds is heterogeneous, which could influence patient and physician choice in case of comparable efficacy. All ADCs show some hematologic and hepatic toxicity; however, DS-8201a, harboring a topoisomerase I inhibitor, showed additional gastrointestinal toxicity, while SYD985 was associated with ocular toxicity. Future phase 2 and phase 3 trials will clarify how these toxicities will influence treatment intensity and adherence.

HER2-directed ADCs are characterized by a high drug-to-antibody ratio and these ADCs promote a bystander killing effect, which explain the antitumor activity even in HER2 low expressing tumors. A higher drug-to-antibody ratio increases the amount of internalized cytotoxic drug molecules despite a low HER2 antigen density on the tumor cell surface. As HER2 is not only expressed on tumor cells but also physiologically on cell membranes of epithelial cells, heart and skeletal muscle cells, the therapeutic window of these investigational ADCs might be challenging [3,4].

Preclinical and early clinical efficacy data of DS-8201a, SYD985, MEDI4276 and XMT-1522 in HER2-low breast cancers are of special interest. About 50% of breast cancers can be categorized as HER2-low and the availability of a targeted treatment option for this patient population would be of great interest. This is especially true for patients with triple-negative breast cancer, the breast cancer subtype with the worst prognosis with a high clinical demand for further therapeutic options.

## Figures and Tables

**Figure 1 ijms-20-01115-f001:**
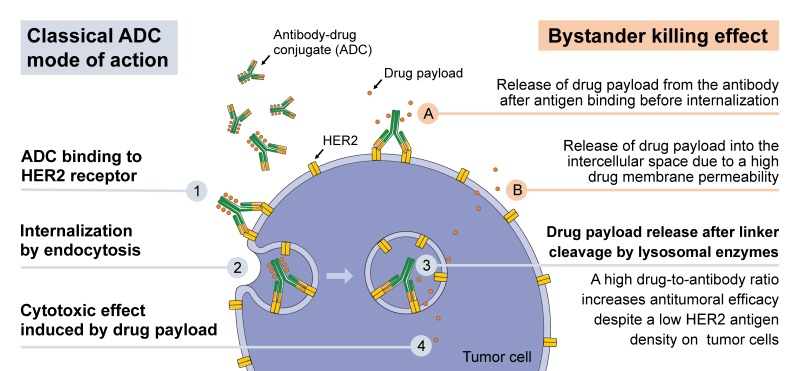
Mode of action of HER2 directed ADCs in HER2-low tumors. Classical mode of action of ADCs with cleavable linkers: (1) After binding of the monoclonal anti-HER antibody component to HER2 expressed on the cell surface of tumor cells, (2) the ADC-HER2 complex is internalized by endocytosis. (3) After linker cleavage by lysosomal proteases, the drug payload is released and (4) can induce the cytotoxic effect leading to tumor cell death. A high drug-to-antibody ratio can increase antitumoral efficacy despite a low HER2 antigen density on tumor cells. Bystander killing effect: Using cleavable linkers, ADCs can be designed to promote drug release from the target cell to the extracellular space. Thereby, surrounding and bystander cells, which may or may not express the ADC target antigen, can be killed by taking up the cytotoxic drug. (A) This bystander killing can occur if the cytotoxic drug is released from the antibody after antigen binding before internalization. (B) Additionally, the drug payload can be released from the tumor cell into the intracellular space due to a high membrane-permeability of the ADC drug payload. This figure was created using Servier Medical Art templates, which are licensed under a Creative Commons Attribution 3.0 Unported License; https://smart.servier.com.

**Table 1 ijms-20-01115-t001:** Investigational HER2 targeting antibody drug conjugates in breast cancer.

Drug Name	Cytotoxic Payload	Reported Efficacy in HER2-Low	Phases (Number of Trials, NCT Identifier)	Company
A166	NA	no	Phase 1/2: 1 (NCT03602079)	Klus Pharma, Inc.
ALT-P7 (HM2-MMAE)	monomethyl auristatin E	no	2: 1 (NCT03281824)	Alteogen, Inc.
ARX788	monomethyl auristatin F	no	1: 2 (NCT02512237, NCT03255070)	Ambrx, Inc.
DHES0815A (anti-HER2/PBD-MA)	PBD-MA	no	1: 1 (NCT03451162)	Genentech, Inc.
DS-8201a (Trastuzumab deruxtecan)	DXd	yes	1: 3 (NCT03523572, NCT03368196, NCT03366428) 2: 1 (NCT03248492) 3: 3 (NCT03734029, NCT03523585, NCT03529110)	Daiichi Sankyo, Inc.
MEDI4276	AZ13599185	yes	-	MedImmune, LLC
RC48	monomethyl auristatin E	no	1b/2: 1 (NCT03052634) 2: 1 (NCT03500380)	RemeGen
SYD985 ([vic-]trastuzumab duocarmazine)	seco-DUBA	yes	3: 1 (NCT03262935)	Synthon Biopharmaceuticals BV
T-DM1 (Trastuzumab emtansine)	DM1	no	1: 3 (NCT02073916, NCT02038010, NCT03364348) 2: 3 (NCT03587740, NCT02073487, NCT02414646)	Roche
XMT-1522 (TAK-522)	AF-HPA	yes	1: 1 (NCT02952729)	Mersana Therapeutics

(clinicaltrials.gov, last access on 20th of December 2018); NA: not available; PBD-MA: pyrrolo[2,1-c][1,4]benzodiazepine monoamide; seco-DUBA: synthetic duocarmycin analogon seco-DUocarmycin-hydroxyBenzamide-Azaindole; AF-HPA: Auristatin F-hydroxypropylamide.

**Table 2 ijms-20-01115-t002:** Investigated ADCs in breast cancer targeting receptors other than HER2.

Drug	Target	Running Trials (Number of Trials, NCT Identifier)	Company
U3-1402	HER3	Phase 1/2: 1 (NCT02980341)	Daiichi Sankyo, Inc.
SGN-LIV1A	LIV1	Phase 1: 1 (NCT01969643)	Seattle Genetics, Inc.
CAB-ROR2-ADC	ROR2	Phase 1/2: 1 (NCT03504488)	BioAtla, LLC
Sacituzumab govitecan (IMMU-132)	Trop-2	Phase 1/2: 1 (NCT01631552) Phase 2: 1 (NCT02161679)	Immunomedics, Inc.

(clinicaltrials.gov, last access on 20th of December 2018; ROR2: Receptor tyrosine kinase-like orphan receptor 2.

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
