# Peer review of "HER2 Directed Antibody-Drug-Conjugates beyond T-DM1 in Breast Cancer"

_ijms, 2019, doi:10.3390/ijms20051115_

Reviewer 1 Report

Authors have done a good job summarizing the ADC targeting HER2 in the review. Thorough review of all available ADCs against HER2 are gathered and presented by the authors, explaining their constituents and briefing about their preclinical and clinical outcomes. However, the benefits and challenges of such ADCs were missing in the discussion. I would recommend adding some discussion about those issues.

Some minor comments:

1.     Many statements and results are missing references. I recommend adding reference to each new result even though they are from the same study that you have referenced previously.

2.     There is a typo on Figure 1 text, “effic cy”

3.     A thorough revision of grammar and sentence structure of the manuscript is required. I would recommend avoiding long sentences (3-4 lines) with many commas, it reduces readability. Some of the sentences are incomplete or lacking verbs. Some examples of incorrect grammar or sentence structures are, lines 280-281, 283-285, 291-292, 308-310, 380-382 etc.

4.     Often ineffectiveness is misinterpreted as inactivity. Compounds are biologically defined with effectiveness whereas chemically defined by activity. I guess author meant “ineffective” in sentence 440 instead of “inactive”.

5.     Be consistent in defining xenograft mouse model, transplant or implant or inoculation.

6.     The sub heading “Drug structure” is misleading since no chemical structures of compounds are illustrated there. Only the ADC constituents are defined, hence “ADC constituents” would be more appropriate heading.

Author Response

Thank you for your comments regarding our manuscript. We have revised our manuscript according to your valuable input: 

Reviewer 1: However, the benefits and challenges of such ADCs were missing in the discussion:

Following paragraphs have been added to our manuscript:

Lines 514-517: “HER2 is a rational target for ADCs in breast cancer as about 15% of breast cancers show HER2 protein overexpression and/or HER2 gene amplification [6], and further 50% show a low-level expression of HER2 without HER2 amplification [9,10]. Additionally, HER2 testing in breast cancer is standardized by international guidelines and reporting of the HER2 status mandatory in clinical routine [9].”

Lines 533-538: “HER2 directed ADCs are characterized by a high drug-to-antibody ratio and these ADCs promote a bystander killing effect, which explain the antitumor activity even in HER2 low expressing tumors. A higher drug-to-antibody ratio increases the amount of internalized cytotoxic drug molecules despite a low HER2 antigen density on the tumor cell surface. As HER2 is not only expressed on tumor cells but also physiologically on cell membranes of epithelial cells, heart and skeletal muscle cells, the therapeutic window of these investigational ADCs might be challenging [3,4].”

Reviewer 1 - minor comment 1: Many statements and results are missing references. I recommend adding reference to each new result even though they are from the same study that you have referenced previously:
Missing references have been added (lines 27, 197, 198, 199, 216, 218, 283, 319, 393, 403, 404, 405, 407, 458, 473, 476)

Reviewer 1 - minor comment 2: There is a typo on Figure 1 text, “effic cy”

The typo has been corrected. 

Reviewer 1 - minor comment 3: A thorough revision of grammar and sentence structure of the manuscript is required. I would recommend avoiding long sentences (3-4 lines) with many commas, it reduces readability. Some of the sentences are incomplete or lacking verbs. Some examples of incorrect grammar or sentence structures are, lines 280-281, 283-285, 291-292, 308-310, 380-382 etc.

The grammar and sentence structure have been revised.

Reviewer 1 - minor comment  4: Often ineffectiveness is misinterpreted as inactivity. Compounds are biologically defined with effectiveness whereas chemically defined by activity. I guess author meant “ineffective” in sentence 440 instead of “inactive”.

“Inactive” has been replaced by “ineffective” in lines 461 und 473.

Reviewer 1 - minor comment  5: Be consistent in defining xenograft mouse model, transplant or implant or inoculation.

“Inoculated” has been replaced by “implanted” in lines 330 and 334.

Reviewer 1 - minor comment  6: The sub heading “Drug structure” is misleading since no chemical structures of compounds are illustrated there. Only the ADC constituents are defined, hence “ADC constituents” would be more appropriate heading.

“Drug structure” has been replaced by “ADC constituents” in lines 134, 146, 156, 181, 190, 279, 315, 387, and 451

Reviewer 2 Report

Rinnerthaler et al present a consise, but detailed review of current developments of HER2 antibody-based ADCs being developed after T-DMI. The concept of the review is very interesting  and the text is very well written. In opinion of this reviewer, the manuscript deserves publication as it will be useful to clinicians and researchers, in particular entering the field of the anticancer therapies based on targeted delivery of drugs.

My only remark is below:

The authors state (p.2 line 49) “The addition of trastuzumab, a humanized monoclonal antibody targeting HER2, to conventional adjuvant chemotherapy, however, resulted in a significant and clinically relevant reduction of disease free survival (HR 0.60; 95% confidence 50 interval [CI] 0.50 - 0.71, P < 0.001) and overall survival (HR 0.66; 95% CI 0.57 - 0.77, P < 0.00001) [7]”, while Moja et al [7] state: “The combined HRs for overall survival (OS) and disease-free survival (DFS) significantly favoured the trastuzumab-containing regimens….”

Author Response

Thank you for your kind comments regarding our manuscript.

Reviewer 2 – comment 1: The authors state (p.2 line 49) “The addition of trastuzumab, a humanized monoclonal antibody targeting HER2, to conventional adjuvant chemotherapy, however, resulted in a significant and clinically relevant reduction of disease free survival (HR 0.60; 95% confidence 50 interval [CI] 0.50 - 0.71, P < 0.001) and overall survival (HR 0.66; 95% CI 0.57 - 0.77, P < 0.00001) [7]”, while Moja et al [7] state: “The combined HRs for overall survival (OS) and disease-free survival (DFS) significantly favoured the trastuzumab-containing regimens….”

“… reduction of disease free survival … ” has been corrected to “ … prolongation of disease free survival …” (line 49)